# Hypertension associated with serotonin reuptake inhibitors: A new analysis in the WHO pharmacovigilance database and examination of dose-dependency

**Basile Chrétien** [1], **Andry Rabiaza** [2], **Nishida Kazuki** [1], **Sophie Fedrizzi** [3], **Marion Sassier** [3], **Charles Dolladille** [3,4], **Joachim Alexandre** [3,4], **Xavier Humbert** [2,4]*

1 Center for Advanced Medicine and Clinical Research, Section of Biostatistics, Nagoya University Graduate School of Medicine, Aichi, Japan, 2 Normandie Univ, UNICAEN, Department of general practice, Caen, France, 3 CHU Caen, Pharmacology Department, Caen, France, 4 Normandie Univ, UNICAEN, INSERM U1086 ANTICIPE, Caen, France

☯ These authors contributed equally to this work.

* xavier.humbert@unicaen.fr

## Abstract

### Introduction

Recent literature has reported instances of drug associated with hypertension with serotonin reuptake inhibitors (SRIs). Nonetheless, the association between SRIs and hypertension development is the subject of ongoing debate. It remains uncertain whether this is indicative of a class effect, and if dose-effect exist. To investigate the potential class effect associating SRIs with hypertension reporting, we utilized real-world data from VigiBase®, the World Health Organization (WHO) pharmacovigilance database.

### Methods

We conducted an updated disproportionality analysis within VigiBase® to identify a signal of hypertension reporting with individual SRIs by calculating adjusted reporting odds ratios (aRORs) within a multivariate case/non-case study design. Additionally, we explored the presence of a dose-effect relationship.

### Results

The database contained 13,682 reports of SRI associated with hypertension (2.2%), predominantly in women (70.0%). Hypertension was most reported in the 45-64 years old age group (44.8%). A total of 3,879 cases were associated with sertraline, 2,862 with fluoxetine, 2,516 with citalopram, 2,586 with escitalopram, 2,441 with paroxetine, 201 with fluvoxamine and 8 with zimeldine.

A significant ROR was observed for all SRIs in both univariate (RORs ranging from 1.39 to 1.54) and multivariable analyses (aRORs ranging from 1.16 to 1.40) after adjustments for age group, sex, concurrent antihypertensive medication and drugs knowns to induce hypertension, except for fluvoxamine and zimeldine. No dose-response relationship was identified.

**Data availability statement:** Data cannot be shared publicly because of membership of the VIGIBASE database. Data are available from the Uppsala Monitoring Centre (contact via https://who-umc.org/vigibase/) for researchers who meet the criteria for access to confidential data. The data underlying the results presented in the study are available from Uppsala Monitoring Centre (https://who-umc.org/vigibase/).

**Funding:** The author(s) received no specific funding for this work.

**Competing interests:** The authors have declared that no competing interests exist.

**Abbreviations:** ADR, Adverse Drug Reaction; aROR, adjusted Reporting Odds Ratio; ATC, Anatomical Therapeutic and Clinical classification; CVD, Cardiovascular Disease; FAERS, FDA Adverse Event Reporting System, ICSR, Individual Case Safety Report; MAOI, Monoamine Oxidase Inhibitors; ROR, Reporting Odds Ratio; SMQ, Standardized MedDRA Query; SPC, Summary of Product Characteristics; SRI, Serotonin Reuptake Inhibitor; WHO, World Health Organization.CIConfidence Interval

## Conclusion

This investigation, conducted under real life conditions, unveils a notable pharmacovigilance safety signal associating SRI usage with hypertension reporting. No dose-response effect was detectable. Further longitudinal studies are warranted.

## Introduction

Hypertension significantly contributes as a risk factor for cardiovascular diseases (CVD) [1,2]. Although essential hypertension predominates clinically, secondary hypertension can emanate from conditions such as renal parenchymal disease, renal artery stenosis, hyperaldosteronism, or pheochromocytoma. Pharmacologically induced hypertension, though less acknowledged, can lead to induced or uncontrolled hypertension [3].

A case-control study involving 700 participants across two hospitals revealed that a history of depression or anxiety correlated with a heightened risk of uncontrolled hypertension (adjusted OR 1.82, 95% CI: 1.27-2.60) [4]. Furthermore, the conjecture that antidepressants could significantly influence blood pressure—either directly or by moderating depression—has emerged [5]. Monoamine oxidase inhibitors (MAOIs), for example, can precipitate severe hypertension in the presence of tyramine-rich food or amphetamine consumption through the potentiation of monoamines like norepinephrine [6]. Tricyclic antidepressants, in a cohort study of 2,981 subjects, manifested a greater likelihood of developing stage 1 hypertension (OR 1.90, 95% CI 0.94-3.84, p < 0.07) and stage 2 hypertension (OR 3.19, 95% CI 1.35-7.55, p < 0.01) [7]. However, tricyclic antidepressant only showed a trend toward grade 1 hypertension in this paper, as the threshold of statistical significance was not reached. These compounds could also induce hypertensive crises in patients with unrecognized pheochromocytoma [8,9]. Consequently, serotonin reuptake inhibitors (SRIs) are touted as first-line antidepressants, given their comparatively lower risk profile for adverse drug reactions (ADRs) [10]. Nonetheless, SRI-associated hypertension debate persists, and hypertension is not universally acknowledged as a possible adverse effect in the Summaries of Product Characteristics (SPC) of this drug class. Hence, following prior indications of a pharmacovigilance signal for SRI associated hypertension in VigiBase, we now conduct an updated investigation with enhanced statistical power and refined methodology accounting for potential confounders and seek to elucidate any dosage-dependent association. This is imperative for advancing patient care optimization [11].

To investigate the potential class effect associating SRIs with hypertension reporting, we utilized real-world data from VigiBase®, the World Health Organization (WHO) pharmacovigilance database and to determine if a dose-effect association was found with SRI- hypertension couples.

## Materials and methods

### Population/Data source

Our analyses harnessed VigiBase®, the WHO's comprehensive Individual Case Safety Report (ICSR) database [12]. Encompassing more than 35 million reports from 120 nations since 1968, ICSRs integrate administrative data (reporting country, report type, reporter qualifications), patient demographics (sex, age), reaction onset date, outcome characterization using MedDRA version 27.1 terms, WHO causality assessment, and detailed drug information (name, administration and cessation dates, induction period, prescribed indication, dosage, dechallenge, rechallenge), accompanied by report completeness levels. Each ADR was classed as either

'serious' or 'non-serious' based on WHO criteria, with 'serious' encompassing outcomes such as death, life-threatening situations, hospitalization or its prolongation, persistent incapacity or disability, and clinically significant events as judged by the reporting physician [13].

### Case/Non-case study in VigiBase®

ICSRs with reported hypertension—based on MedDRA *System Organ Class version* 27.1's "Hypertension (broad)" Standardized MedDRA Query (SMQ) definition—from 1968 to the 29th of July 2024 comprised our 'cases' within the adult demographic, while 'non-cases' involved all adult reports devoid of hypertension. Terms included in the "Hypertension (broad)" (SMQ) are described in detail in Supplementary S3 Table. We classified drug exposure by detecting at least one SRI as per the Anatomical Therapeutic and Clinical classification (ATC) within a report. All SRI were included in the analysis regardless of their status (suspect, concomitant or interaction). The analysis of disproportionality in this case/non-case format, aimed to examine the association between drug exposure and hypertension occurrence [14–16], facilitated the estimation of reporting odds ratios (RORs) and their 95% confidence intervals (CIs) for each SRI.

### Statistical analysis

Data were expressed as means ± SD or percentage. For the case/non-case analysis, we estimated the ROR with 95% CI for each SRI-hypertension pair, and a safety signal was identified with an ROR > 1 and a lower 95% CI limit > 1. Furthermore, multivariable analyses were performed, adjusting for age, sex, concurrent C02 ATC class antihypertensive reports, and drugs known to induced hypertension as defined by Foy et al. [17] to reduce bias using adjusted reporting odds ratio (aRORs). The investigation of dose-response relationships involved ROR computations for dosages less than or equal to the median and those exceeding it for each SRI, with a linear regression analysis conducted likewise. Time to onset, number of positives dechallenges and rechallenges were also described. A sensitivity analysis was done using only reports with suspected SRIs (according to the pharmacovigilance criteria). Another sensitivity analyses was done in a population of patients treated with antidepressants (defined by the N06A ATC class). The analyses encompassed the collective SRIs and adhered to the READUS-PV guidelines with an execution in R 4.4.1 [18,19].

## Results

### Descriptive characteristics

VigiBase® reports pertinent to the included SRIs totaled 625,205. Adhering to the selection criteria, we identified 13,688 hypertensive cases (2.2%) (Fig 1).

SRI associated with hypertension predominately affected women (70.0%). Hypertension was most reported in the 45-64 years old age group (44.8%). Reports associated with sertraline numbered at 3,879; fluoxetine, 2,862; citalopram, 2,516; escitalopram, 2,586; paroxetine, 2,441; fluvoxamine, 201 and zimeldine, 8. Concomitant antihypertensive drug use was reported in 1,095 cases (8.0%), and a total of 1,752 reports were linked with CVDs (750 myocardial infarction cases and 1,002 stroke cases). Serious cases constituted 9,223 reports (75.5%) (Table 1).

Time to onset and positive dechallenge - rechallenge are described in Table 2 and 3 respectively.

### Univariate and multivariate case/non-case analysis in VigiBase

Table 4 illustrates that SRIs elicited a significant signal for hypertension in bivariate analysis (RORs ranged from 1.43 to 1.58) and multivariable analysis (aRORs ranged from 1.16 to

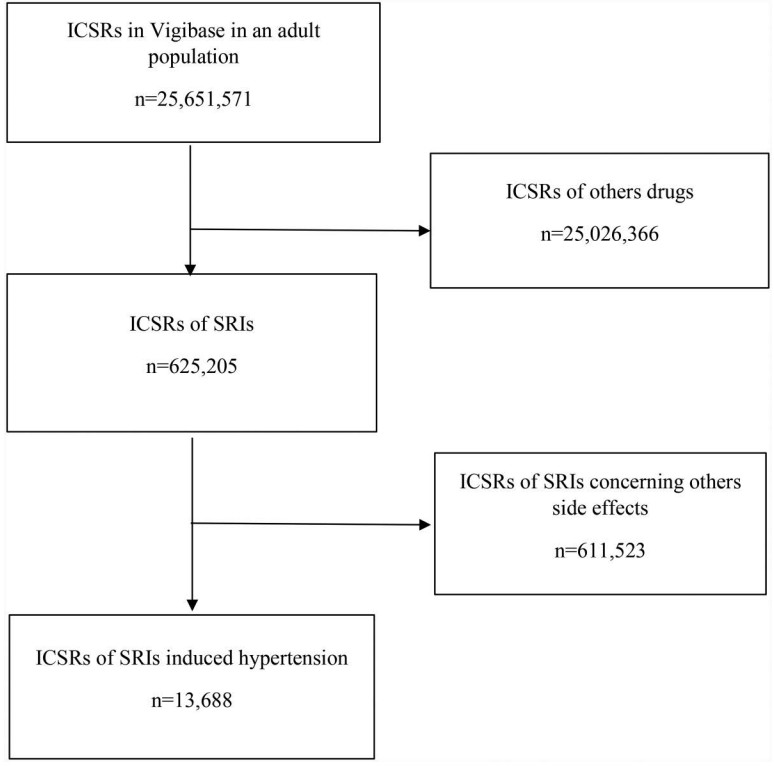

**Fig 1. Flow chart of individual case safety report (ICSR) in WHO pharmacovigilance database concerning serotonin reuptake inhibitors (SRIs) induced hypertension.**

1.40) post-adjustment for age group, sex, concurrent antihypertensive medication and drugs knowns to induce hypertension, except for fluvoxamine and zimeldine.

## Dose-effect SRIs associated with hypertension

No dose-response relationship emerged (Table 5).

## Sensitivity analysis

Disproportionality analysis with all suspected SRIs in VigiBase® to search for a signal of hypertension found no pharmacovigilance signal except for sertraline [RORa = 1.07 (1.00;1.15)] in multivariate analyses (Supplementary S1 Table). In another hand, disproportionality analysis with SRIs in VigiBase® to search for a signal of hypertension in a population of patients treated with antidepressants found a pharmacovigilance signal for citalopram, escitalopram, fluoxetine, paroxetine and sertraline in multivariate analyses (Supplementary S2 Table).

## Discussion

This investigation substantiates a pharmacovigilance signal for hypertension in patients administered SRIs, despite adjustments for multiple confounders. A class effect seems plausible, yet no dose-dependency was identified. Past VigiBase studies on this topic are sparse, potentially with suboptimal methodologies. Our prior work relied solely on univariate analysis, neglected potential confounders, and omitted the search for dose-dependency relationships [11]. Montastruc et al. identified a positive correlation between the Norepinephrine

**Table 1. Characteristics of the 13,688 individual case safety reports (ICSRs) of hypertension with SRIs reported in VigiBase®.**

| Sex | N available | 13,688 |
|---|---|---|
| | **Female** | **9,535 (70.0%)** |
| Age | N available | 13,688 |
| | 18-44 | 3,326 (24.3%) |
| | 45-64 | 6,134 (44.8%) |
| | 65-74 | 2,452 (17.9%) |
| | 75 + | 1,776 (13.0%) |
| Serious | N available | 12,208 |
| | Yes | 9,223 (75.5%) |
| Seriousness criterias* | Caused/Prolonged Hospitalization | 4,287 (46.5%) |
| | Congenital anomaly/Birth defect | 0 |
| | Death | 709 (7.7%) |
| | Disabling/Incapacitating | 247 (2.7%) |
| | Life threatening | 523 (5.7%) |
| | Other | 3,358 (36.4%) |
| | NA's | 4,564 (49.5%) |
| Concomitant stroke | | 1,002 (7.3%) |
| Concomitant myocardial infarction | | 750 (5.5%) |
| Concomitant antihypertensive drug | | 1,095 (8.0%) |
| Concomitant potentially hypertensive drug | | 5,575 (40.7%) |

*: several seriousness criteria can be added at the same time.

**Table 2. Time to onset (TTO) of hypertension after SRIs exposure.**

| Drug | Number of reports with available information | TTO in days (median, interquartile range) |
|---|---|---|
| Alaproclate | 0 | NA |
| Citalopram | 157 | 2.0 (0.0-16.0) |
| Escitalopram | 136 | 1.0 (0.0-12.5) |
| Etoperidone | 0 | NA |
| Fluoxetine | 249 | 13.0 (1.0-66.0) |
| Fluvoxamine | 59 | 9.0 (1.0-38.0) |
| Paroxetine | 310 | 5.0 (1.0-119.0) |
| Sertraline | 388 | 4.5 (0.0-56.0) |
| Zimeldine | 5 | 1.0 (1.0-4.0) |

**Table 3. Dechallenge and rechallenge after the onset of hypertension.**

| Drug | Positive dechallenge | Positive rechallenge |
|---|---|---|
| Alaproclate | 0 | 0 |
| Citalopram | 138 | 3 |
| Escitalopram | 168 | 2 |
| Etoperidone | 0 | 0 |
| Fluoxetine | 253 | 15 |
| Fluvoxamine | 41 | 2 |
| Paroxetine | 212 | 5 |
| Sertraline | 275 | 12 |
| Zimeldine | 1 | 0 |

**Table 4. Disproportionality analysis with all SRIs in VigiBase® to search for a signal of hypertension.**

| Drug Name | Cases (=a) | Non-cases (=b) | Bivariate analysis | | Multivariable analysis* | |
|---|---|---|---|---|---|---|
| | | | ROR | 95%CI | ROR | 95%CI |
| alaproclate | 0 | 1 | NA | | | |
| citalopram | 2,516 | 113,000 | 1.43 | (1.37-1.49) | 1.16 | (1.12-1.21) |
| escitalopram | 2,586 | 109,160 | 1.52 | (1.46-1.58) | 1.25 | (1.20-1.30) |
| etoperidone | 0 | 34 | NA | | | |
| fluoxetine | 2,862 | 125,242 | 1.47 | (1.41-1.52) | 1.29 | (1.25-1.34) |
| fluvoxamine | 201 | 12,673 | 1.01 | (0.88-1.17) | 0.99 | (0.86-1.14) |
| paroxetine | 2,441 | 9,8904 | 1.58 | (1.52-1.65) | 1.40 | (1.34-1.46) |
| sertraline | 3,879 | 165,784 | 1.50 | (1.45-1.55) | 1.30 | (1.26-1.35) |
| zimeldine | 8 | 910 | 0.56 | (0.28-1.13) | 0.59 | (0.29-1.18) |
| SRI class | 13,688 | 610,679 | 1.45 | (1.43-1.48) | 1.33 | (1.31-1.35) |

*adjusted on age category, sex, antihypertensive drugs associated (ATC codes) and drugs known to induce hypertension.

**Table 5. Disproportionality analysis of the signal of hypertension associated with SRIs to search for a dose-dependency effect.**

| Drug Name | Doses ≤ to the median | | | | Doses> to the median | | | | P*² |
|---|---|---|---|---|---|---|---|---|---|
| | Bivariate analysis | | Multivariable analysis*¹ | | Bivariate analysis | | Multivariable analysis*¹ | | |
| | ROR | 95%CI | ROR | 95%CI | ROR | 95%CI | ROR | 95%CI | |
| alaproclate | NA | | | | | | | | |
| citalopram | 1.81 | (1.59-2.07) | 1.16 | (1.12-1.21) | 2.02 | (1.68-2.42) | 1.70 | (1.42-2.04) | 0.70 |
| escitalopram | 1.40 | (1.20-1.64) | 1.25 | (1.20-1.30) | 1.6 | (1.37-1.86) | 1.47 | (1.26-1.72) | 0.71 |
| etoperidone | NA | | | | | | | | |
| fluoxetine | 1.62 | (1.38-1.90) | 1.29 | (1.25-1.34) | 1.76 | (1.45-2.14) | 1.57 | (1.29-1.90) | 0.71 |
| fluvoxamine | 1.17 | (0.58-2.35) | 0.99 | (0.86-1.14) | 1.00 | (0.59-1.69) | 0.93 | (0.54-1.61) | 0.84 |
| paroxetine | 1.70 | (1.45-1.98) | 1.40 | (1.34-1.46) | 2.82 | (2.35-3.39) | 2.51 | (2.09-3.01) | 0.62 |
| sertraline | 1.99 | (1.83-2.18) | 1.30 | (1.26-1.35) | 0.92 | (0.44-1.95) | 0.83 | (0.39-1.75) | 0.58 |
| zimeldine | NA | | | | NA | | | | |

*¹: adjusted on age category, sex, antihypertensive drugs associated (ATC codes) and drugs known to induce hypertension, *²: linear regression model. Model was run only if at least 5 reports of hypertension were made for each drug in the conditions given in this table.

Transporter/Serotonin transporter pKi ratio and hypertension reports associated with SRI and SNRI antidepressants using univariate analysis [20].

Controversy surrounds SRI associated with hypertension, with only sertraline and paroxetine SPCs explicitly noting hypertension risk. These two agents, in pre-clinical and phases II-III trials, were connected to potential hypertension manifestations [21–23], unlike the other four agents [24–29], which may account for this inconsistency. Regrettably, these studies predominantly featured normotensive subjects, which do not perfectly mirror real-life patient treatments. Elsewhere, Chen et al. have realized an evaluation of the association between the six most commonly prescribed SRIs and CVD adverse events, using the FDA Adverse Event Reporting System (FAERS) from Q1 2004 to Q2 2022. Their disproportionality analysis showed a significant association between SRIs and the CVD adverse events (arrhythmias, *torsades de pointes*/QT prolongation, cardiomyopathy, and hypertension), with higher signals in middle-aged and elderly patients and women [30]. However, Zhang et al. have previously shown that SRIs help to reduce peak blood pressure in hypertensive patients with depression in a metanalysis including only six studies, and 149 patients with CVD and depression.

Potential pathophysiological mechanisms may lie in serotonin function, sympathetic activation and genetic heterogeneity [31].

Hypertension mechanisms induced by SRIs might involve the blockade of serotonin transporters, causing increased extracellular monoaminergic neurotransmitter concentrations [32]. In vivo and in vitro evidence suggests that heightened serotonin levels can cause arterial vasoconstriction [33,34] and that SRIs alter serotonin uptake in platelets [35]. Other studies have speculated that SRIs might hamper nitric oxide synthesis, a key vasodilator integral to vascular tone and reactivity [36]. Katsi et al.'s narrative review in 2013 found minimal SRI effects on blood pressure [37]. However, Grossman et al. highlighted fluoxetine associated with hypertension in case studies [38]. Amsterdam et al. observed a marginal incidence of sustained hypertension (1.7%) during short-term fluoxetine treatment at a daily dosage of 20mg over up to 12 weeks in 796 depressed patients [39]. Experimental investigations also demonstrated fluoxetine's potential to raise blood pressure under various conditions (e.g., in ambulatory rats, dogs with neurogenic orthostatic hypotension, and individuals with idiopathic Parkinson's disease) [40–42]. Montastruc et al. noted a mild yet clinically significant impact of fluoxetine on hemodynamic factors, including blood pressure, in parkinsonian patients with orthostatic hypotension [41]. From a pathophysiological point of view, these same authors have suggested that the NET/SERT ratio could be involved, although they tend to favour the determining role of NET over the serotonergic effect [20]. Furthermore, an orthostatic hypotension-simulating tilt test in chronically sino-aortic denervated dogs under chloralose anaesthesia yielded analogous results [40]. Lazartigues et al. discovered that acute central fluoxetine administration induced a pressor response via enhancement of sympathetic tone and vasopressin secretion, hinting at the potential applicability of $\alpha_1$-adrenoceptor and/or $V_{1A}$-vasopressin receptor antagonists in managing 'Serotonin Syndrome' [42]. Research on hypertension induced by other SRIs remains scant.

Highlighted strengths of this study include the utilization of the validated case/non-case methodology for ADR signal detection in pharmacovigilance data [15,16]. VigiBase®, with its global report aggregation, offers diverse medical practices and patient characteristics. The enhanced methodology, featuring multivariable analysis, confers greater relevance to this latest analysis. Notwithstanding, limitations common to pharmacovigilance database investigations persist, such as the prominence of under-reporting, which, while important, does not detract from the validity and significance of the case/non-case methodology [15]. Missing data also represents a limitation in pharmacovigilance database extractions. Additionally, given depression's potential impact on blood pressure, discerning the precise influence of SRIs on blood pressure remains challenging. However, given the absence of signal of hypertension in the analysis including only suspected SRIs, we cannot exclude with certainty that the signal found in our main analysis could be driven by unidentified confounding factors. It is indeed possible that a part of our signal is driven by the fact that depressive symptoms are associated with a greater risk of developing hypertension [43]. Furthermore, the analysis comparing SRIs to other antidepressants indicates that SRIs were associated with a lower signal for hypertension. However, this finding might be influenced by a notoriety bias favoring these antidepressants, which ultimately makes our hypothesis plausible. The short time to onset of hypertension after SRIs initiation, as well as the number of positive dechallenges and rechallenges are also in favour of our hypothesis.

## Conclusions

This updated analysis, carried out under real-world circumstances, has disclosed a significant pharmacovigilance safety signal linking SRI administration with the onset or exacerbation of hypertension. From a clinical perspective, this study implies that physicians treating patients

with depressive disorders must recognize the potential for SRIs to elevate blood pressure akin to other antidepressants. Consequently, routine blood pressure monitoring is advisable for these patients. Given the absence of a dose-effect relationship, clinicians should determine SRI dosing based on symptomatology and observed guidelines.

## Supporting information

**S1 Table. Disproportionality analysis with all suspected SRIs in VigiBase® to search for a signal of hypertension.** *adjusted on age category, sex, antihypertensive drugs associated (ATC codes) and associated drugs known to induce hypertension (ATC codes).
(DOCX)

**S2 Table. Disproportionality analysis with SRIs in VigiBase® to search for a signal of hypertension in a population of patients treated with antidepressants.** *adjusted on age category, sex, antihypertensive drugs associated (ATC codes) and associated drugs know to induce hypertension (ATC codes). Note: the population was defined using N06A ATC codes (antidepressant), and not with International Non Proprietary Names, which explains the slight difference in the number of cases between this sensitivity analyses and the main one.
(DOCX)

**S3 Table. Terms included in the SMQ "Hypertension (broad)" definition based on Med-DRA SOC.**
(DOCX)

## Author contributions

**Conceptualization:** Basile Chrétien, Marion Sassier, Charles Dolladille, Xavier Humbert.

**Formal analysis:** Xavier Humbert.

**Methodology:** Basile Chrétien, Xavier Humbert.

**Supervision:** Xavier Humbert.

**Validation:** Xavier Humbert.

**Writing – original draft:** Basile Chrétien, Marion Sassier, Joachim Alexandre, Xavier Humbert.

**Writing – review & editing:** Andry Rabiaza, Nishida Kazuki, Sophie Fedrizzi, Charles Dolladille.

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
