## [Decision Letter · Decision Letter 0]

22 Oct 2024

PONE-D-24-22756Hypertension associated with serotonin reuptake inhibitors: an updated analysis in the WHO pharmacovigilance database and examination of dose-dependencyPLOS ONE

Dear Dr. Humbert,

Thank you for submitting your manuscript to PLOS ONE. After careful consideration, we feel that it has merit but does not fully meet PLOS ONE’s publication criteria as it currently stands. Therefore, we invite you to submit a revised version of the manuscript that addresses the points raised during the review process.

We look forward to receiving your revised manuscript.

Kind regards,

Sheikh Arslan Sehgal, PhD

Academic Editor

PLOS ONE

2. We note that you have indicated that there are restrictions to data sharing for this study. For studies involving human research participant data or other sensitive data, we encourage authors to share de-identified or anonymized data. However, when data cannot be publicly shared for ethical reasons, we allow authors to make their data sets available upon request. For information on unacceptable data access restrictions, please see http://journals.plos.org/plosone/s/data-availability#loc-unacceptable-data-access-restrictions. Before we proceed with your manuscript, please address the following prompts: a) If there are ethical or legal restrictions on sharing a de-identified data set, please explain them in detail (e.g., data contain potentially identifying or sensitive patient information, data are owned by a third-party organization, etc.) and who has imposed them (e.g., a Research Ethics Committee or Institutional Review Board, etc.). Please also provide contact information for a data access committee, ethics committee, or other institutional body to which data requests may be sent. b) If there are no restrictions, please upload the minimal anonymized data set necessary to replicate your study findings to a stable, public repository and provide us with the relevant URLs, DOIs, or accession numbers. Please see http://www.bmj.com/content/340/bmj.c181.long for guidelines on how to de-identify and prepare clinical data for publication. For a list of recommended repositories, please see https://journals.plos.org/plosone/s/recommended-repositories. You also have the option of uploading the data as Supporting Information files, but we would recommend depositing data directly to a data repository if possible. Please update your Data Availability statement in the submission form accordingly.

3. Thank you for uploading your study's underlying data set. Unfortunately, the repository you have noted in your Data Availability statement does not qualify as an acceptable data repository according to PLOS's standards. At this time, please upload the minimal data set necessary to replicate your study's findings to a stable, public repository (such as figshare or Dryad) and provide us with the relevant URLs, DOIs, or accession numbers that may be used to access these data. For a list of recommended repositories and additional information on PLOS standards for data deposition, please see https://journals.plos.org/plosone/s/recommended-repositories.

Additional Editor Comments (if provided):

Reviewers' comments:

Reviewer's Responses to Questions

**Comments to the Author**

1. Is the manuscript technically sound, and do the data support the conclusions?

Reviewer #1: Yes

Reviewer #2: Yes

2. Has the statistical analysis been performed appropriately and rigorously? 

Reviewer #1: Yes

Reviewer #2: Yes

3. Have the authors made all data underlying the findings in their manuscript fully available?

Reviewer #1: Yes

Reviewer #2: Yes

4. Is the manuscript presented in an intelligible fashion and written in standard English?

Reviewer #1: Yes

Reviewer #2: Yes

5. Review Comments to the Author

Reviewer #1: Thank you for the opportunity given to review this work.

I appreciate the study conducted by the authors as the findings bring new insight to the specific topic. Please find below some suggestions to add to the quality of this paper regarding clarification.

TITLE/ABSTRACT

Comment 1: the word “an updated” seems a bit ambiguous without clear information given.

Comment 2: “..predominantly in women (69.9%) within the age range of 45-64 years (45.2%).” Is the proportion of 45.2% for females only?. It seems there are two different information that should not be combined in this sentence.

MATERIALS AND METHODS/RESULTS/DISCUSSION

Comment 3: “Hypertension (broad)” definition based on MedDRA SOC is used by the authors to identify cases of interest. Does this selection include gestational hypertension cases as well?. Additional information is suggested to clarify this.

Comment 4: Did the authors consider the positive dechallenge or rechallenge given in the reports when selecting hypertensive cases?. Is there a preference for drug characterisation by reporters (for example only reports where an SRI drug is suspected) when including the cases for the analyses?

Comment 5: There are also other drugs reported to (possibly) induce hypertension (https://doi.org/10.1016/j.ecl.2019.08.013). Supposedly these drugs are also mentioned in the reports together with SRI drugs of interest, the chance for hypertension may increase because of potential drug interactions. Did the authors take this possibility into account?.

Comment 6: Could the authors add additional information if there is a plausible overlapping between the timing use of an SRI and the onset of hypertension mentioned in the reports?.

Reviewer #2: The authors tackle a question that is still unresolved in pharmacovigilance: the risk of hypertension (HTN) in patients treated with SSRIs. They rely on a classic but well-established methodology of disproportionality, partially multivariate, and have rightly followed the recent READUS-PV guidelines.

Methodologically, it seems necessary to clarify the inclusion criteria for cases. Based on the numbers provided by the authors, it appears that they have chosen to include all cases mentioning an SSRI (suspect OR concomitant), not just the cases where the SSRI was considered suspected. This choice is not necessarily absurd, but it must be clearly stated in the methodology. The analysis would benefit from a sensitivity analysis restricted to cases where SSRIs are suspected (excluding concomitant cases). I’m not sure that disproportionality remains positive with these criteria, according to the tests I conducted on the database.

To strengthen their results, the authors might consider comparing the effect of SSRIs on blood pressure with that produced by other medications. It would be very interesting to compare SSRIs to another medication used in a similar population (depressed patients) to mitigate some of the confounding biases. However, the challenge is that most antidepressants are more or less strongly associated with the occurrence of HTN (tricyclics, MAO inhibitors, SNRIs…). As a fallback, perhaps a comparison of SSRIs vs all other reported antidepressants and/or SSRIs vs SNRIs and/or SSRIs vs mirtazapine/mianserine could be proposed. This would better account for the ROR.

The results section is somewhat brief, even though it refers to high-quality tables. It could potentially be slightly expanded to balance the article.

The scientific literature is relatively rich in studies that contribute to the debate on the HTN-SSRI link. This debate might warrant a more precise discussion in the article. In particular:

- Reference 7 on tricyclics only shows a trend toward grade 1 HTN (lower bound of the OR <1), which should be clarified in the introduction.

- The following article deserves to be discussed (in the introduction or discussion), especially since it seems rather reassuring regarding the effect of SSRIs on blood pressure (with even a decrease in diastolic pressure observed): https://doi.org/10.1016/j.jad.2023.08.032

- The following article could also be mentioned, even if it is more consistent with the authors’ theory in the manuscript: https://doi.org/10.1016/j.psychres.2023.115300

- From a pathophysiological perspective, the hypothesis by Montastruc et al. regarding the NET/SERT ratio is interesting to mention, even if it leans more toward the determining role of NET compared to the serotonergic effect.

One minor point: it seems that there is a missing arrow in Fig 1 (flowchart).

Overall, this is a very interesting article that deserves some adjustments to become fully convincing.

6. PLOS authors have the option to publish the peer review history of their article (what does this mean? ). If published, this will include your full peer review and any attached files.

**Do you want your identity to be public for this peer review?** For information about this choice, including consent withdrawal, please see our Privacy Policy .

Reviewer #1: No

Reviewer #2: **Yes: ** Alexandre Gérard

---

## [Author Response · Author response to Decision Letter 1]

30 Dec 2024

PONE-D-24-22756 – R1: Hypertension associated with serotonin reuptake inhibitors: an updated analysis in the WHO pharmacovigilance database and examination of dose-dependency

Reviewer #1:

Thank you for the opportunity given to review this work. I appreciate the study conducted by the authors as the findings bring new insight to the specific topic. Please find below some suggestions to add to the quality of this paper regarding clarification.

REPLY: We appreciate the straightforward synthetized message that this Expert. We thank her/him for the insightful comments raised. In the revised MS all changes introduced are outlined in yellow. Pages and lines indicated in brackets at the end of our comments in this document are taken from the clean manuscript. Finally, we updated the data from our study.

TITLE/ABSTRACT

Comment 1: the word “an updated” seems a bit ambiguous without clear information given.

REPLY: Thank you for this important message. We propose this new title : “Hypertension associated with serotonin reuptake inhibitors: a new analysis in the WHO pharmacovigilance database and examination of dose-dependency” (page 1, line 1).

Comment 2: “..predominantly in women (69.9%) within the age range of 45-64 years (45.2%).” Is the proportion of 45.2% for females only?. It seems there are two different information that should not be combined in this sentence.

REPLY: Thank you for this important message. We have rephrased this sentence (page 2, line 44).

MATERIALS AND METHODS/RESULTS/DISCUSSION

Comment 3: “Hypertension (broad)” definition based on MedDRA SOC is used by the authors to identify cases of interest. Does this selection include gestational hypertension cases as well?. Additional information is suggested to clarify this.

REPLY: Thank you for this important message. We have precisely described “Hypertension (broad)” definition based on MedDRA SOC in Supplementary Table III and this definition includes gestational hypertension. We have precised this point in the Method section (page 7, lines 125-126).

Comment 4: Did the authors consider the positive dechallenge or rechallenge given in the reports when selecting hypertensive cases?. Is there a preference for drug characterisation by reporters (for example only reports where an SRI drug is suspected) when including the cases for the analyses?

REPLY: Thank you for this important message. Positive dechallenge and rechallenge are now described in Table V and a reminder is given in Results section (page 9, line 159). Moreover, All SRIs reported, regardless of their status (suspect, concomitant or interaction), were included in the analyses, as is it standard practice in pharmacoepidemiological analyses. We have precised this point in the Method section (page 7, lines 127-128).

Comment 5: There are also other drugs reported to (possibly) induce hypertension (https://doi.org/10.1016/j.ecl.2019.08.013). Supposedly these drugs are also mentioned in the reports together with SRI drugs of interest, the chance for hypertension may increase because of potential drug interactions. Did the authors take this possibility into account?

REPLY: Thank you for this important message. Now, in the multivariate analysis, drugs known to cause hypertension have been included in the model (from ATC codes) in addition to age category, sex, antihypertensive drugs. This point is precised in Method section (page 8, lines 137-138).

Comment 6: Could the authors add additional information if there is a plausible overlapping between the timing use of an SRI and the onset of hypertension mentioned in the reports?

REPLY: Thank you for this important message. Now, Table IV summarises, for each SRI, the time to onset of hypertension after their introduction, using the median and interquartile ranges. We have precised this point in the Method section (page 8, lines 141-142).

Reviewer #2:

The authors tackle a question that is still unresolved in pharmacovigilance: the risk of hypertension (HTN) in patients treated with SSRIs. They rely on a classic but well-established methodology of disproportionality, partially multivariate, and have rightly followed the recent READUS-PV guidelines.

REPLY: We appreciate the straightforward synthetized message of this Expert. We thank her/him for the insightful comments raised. In the revised MS all changes introduced are outlined in yellow. Pages and lines indicated in brackets at the end of our comments in this document are taken from the clean manuscript. Finally, we updated the data from our study

Methodologically, it seems necessary to clarify the inclusion criteria for cases. Based on the numbers provided by the authors, it appears that they have chosen to include all cases mentioning an SSRI (suspect OR concomitant), not just the cases where the SSRI was considered suspected. This choice is not necessarily absurd, but it must be clearly stated in the methodology.

REPLY: Thank you for this important message. All SRIs reported, regardless of their status (suspect, concomitant or interaction), were included in the analyses. We have precised this point in the Method section (page 7, lines 127-128).

The analysis would benefit from a sensitivity analysis restricted to cases where SSRIs are suspected (excluding concomitant cases). I’m not sure that disproportionality remains positive with these criteria, according to the tests I conducted on the database.

REPLY: Thank you for this important message. We have conduced sensitivity analysis as advised by reviewer. These results are now in Results section (page 9, lines 171-173) and Supplementary Table I.

To strengthen their results, the authors might consider comparing the effect of SSRIs on blood pressure with that produced by other medications. It would be very interesting to compare SSRIs to another medication used in a similar population (depressed patients) to mitigate some of the confounding biases. However, the challenge is that most antidepressants are more or less strongly associated with the occurrence of HTN (tricyclics, MAO inhibitors, SNRIs…). As a fallback, perhaps a comparison of SSRIs vs all other reported antidepressants and/or SSRIs vs SNRIs and/or SSRIs vs mirtazapine/mianserine could be proposed. This would better account for the ROR.

REPLY: Thank you for this important message. We have conduced sensitivity analysis as advised by reviewer. These results are now in Results section (pages 9-10 , lines 173-176) and Supplementary Table II.

The results section is somewhat brief, even though it refers to high-quality tables. It could potentially be slightly expanded to balance the article.

REPLY: Thank you for this important message. We have expanded the Results section to include the results of sensitivity analyses (pages 9-10, lines 171-176).

The scientific literature is relatively rich in studies that contribute to the debate on the HTN-SSRI link. This debate might warrant a more precise discussion in the article. In particular:

- Reference 7 on tricyclics only shows a trend toward grade 1 HTN (lower bound of the OR <1), which should be clarified in the introduction.

REPLY: Thank you for this important message. We have precised this point in Introduction section (page 5, lines 90-91).

- The following article deserves to be discussed (in the introduction or discussion), especially since it seems rather reassuring regarding the effect of SSRIs on blood pressure (with even a decrease in diastolic pressure observed): https://doi.org/10.1016/j.jad.2023.08.032

REPLY: Thank you for this important message. We have precised this point in Discussion section (page 11, lines 197-201).

- The following article could also be mentioned, even if it is more consistent with the authors’ theory in the manuscript: https://doi.org/10.1016/j.psychres.2023.115300

REPLY: Thank you for this important message. We have precised this point in Discussion section (pages 10-11, lines 192-197).

- From a pathophysiological perspective, the hypothesis by Montastruc et al. regarding the NET/SERT ratio is interesting to mention, even if it leans more toward the determining role of NET compared to the serotonergic effect.

REPLY: Thank you for this important message. We have precised this point in Discussion section (page 12, lines 217-219).

One minor point: it seems that there is a missing arrow in Fig 1 (flowchart).

REPLY: Thank you for this message. It has been added.

Overall, this is a very interesting article that deserves some adjustments to become fully convincing.

REPLY: Thank you for your help in improving the quality of our article.

---

## [Decision Letter · Decision Letter 1]

7 Jan 2025

Hypertension associated with serotonin reuptake inhibitors: a new analysis in the WHO pharmacovigilance database and examination of dose-dependency

PONE-D-24-22756R1

Dear Dr. Humbert,

We’re pleased to inform you that your manuscript has been judged scientifically suitable for publication and will be formally accepted for publication once it meets all outstanding technical requirements.

Kind regards,

Sheikh Arslan Sehgal, PhD

Academic Editor

PLOS ONE

Additional Editor Comments (optional):

Reviewers' comments:

Reviewer's Responses to Questions

**Comments to the Author**

1. If the authors have adequately addressed your comments raised in a previous round of review and you feel that this manuscript is now acceptable for publication, you may indicate that here to bypass the “Comments to the Author” section, enter your conflict of interest statement in the “Confidential to Editor” section, and submit your "Accept" recommendation.

Reviewer #2: All comments have been addressed

2. Is the manuscript technically sound, and do the data support the conclusions?

Reviewer #2: Yes

3. Has the statistical analysis been performed appropriately and rigorously? 

Reviewer #2: Yes

4. Have the authors made all data underlying the findings in their manuscript fully available?

Reviewer #2: No

5. Is the manuscript presented in an intelligible fashion and written in standard English?

Reviewer #2: Yes

6. Review Comments to the Author

Reviewer #2: All comments have been addressed.

7. PLOS authors have the option to publish the peer review history of their article (what does this mean? ). If published, this will include your full peer review and any attached files.

**Do you want your identity to be public for this peer review?** For information about this choice, including consent withdrawal, please see our Privacy Policy .

Reviewer #2: **Yes: ** Alexandre O. Gérard

---

## [Editor Report · Acceptance letter]

PONE-D-24-22756R1

PLOS ONE

Dear Dr. Humbert,

I'm pleased to inform you that your manuscript has been deemed suitable for publication in PLOS ONE. Congratulations! Your manuscript is now being handed over to our production team.

Kind regards,

on behalf of

Dr Sheikh Arslan Sehgal

Academic Editor

PLOS ONE